# An evolutionarily conserved iron-sulfur cluster underlies redox sensory function of the Chloroplast Sensor Kinase

Iskander M. Ibrahim [1], Huan Wu[1], Roman Ezhov[2], Gilbert E. Kayanja[1], Stanislav D. Zakharov[3], Yanyan Du[1,4], Weiguo Andy Tao[1], Yulia Pushkar[2], William A. Cramer[3] & Sujith Puthiyaveetil [1]*

Photosynthetic efficiency depends on equal light energy conversion by two spectrally distinct, serially-connected photosystems. The redox state of the plastoquinone pool, located between the two photosystems, is a key regulatory signal that initiates acclimatory changes in the relative abundance of photosystems. The Chloroplast Sensor Kinase (CSK) links the plastoquinone redox signal with photosystem gene expression but the mechanism by which it monitors the plastoquinone redox state is unclear. Here we show that the purified *Arabidopsis* and *Phaeodactylum* CSK and the cyanobacterial CSK homologue, Histidine kinase 2 (Hik2), are iron-sulfur proteins. The Fe-S cluster of CSK is further revealed to be a high potential redox-responsive [3Fe-4S] center. CSK responds to redox agents with reduced plastoquinone suppressing its autokinase activity. Redox changes within the CSK iron-sulfur cluster translate into conformational changes in the protein fold. These results provide key insights into redox signal perception and propagation by the CSK-based chloroplast two-component system.

---

[1] Department of Biochemistry and Center for Plant Biology, Purdue University, West Lafayette, IN 47907, USA. [2] Department of Physics and Astronomy, Purdue University, 525 Northwestern Avenue, West Lafayette, IN 47907, USA. [3] Department of Biological Sciences, Purdue University, West Lafayette, IN 47907, USA. [4] Present address: Shanghai Center for Plant Stress Biology, CAS Center for Excellence in Molecular Plant Sciences, Chinese Academy of Sciences, 200032 Shanghai, China. *email: spveetil@purdue.edu

ron–sulfur (Fe-S) clusters are evolutionarily ancient, ubiquitous, functionally diverse prosthetic groups. Their mid-range redox potential makes them an ideal redox mediator in biological electron transport, such that they are a major component of photosynthetic and the respiratory electron transport chains[1–3]. Fe and S can occur at different stoichiometries, forming diverse Fe-S clusters than span a range of oxidation states. This remarkable versatility in the redox chemistry of the Fe-S cluster is also exploited by several regulatory proteins[4–6], which use them as cofactors for sensing and responding to the intracellular and extracellular redox environment.

Light reactions of oxygenic photosynthesis take place in the thylakoid membrane of chloroplasts and cyanobacteria. Two functionally and spectrally distinct photosystems, PS II and PS I, are connected in series in the photosynthetic electron transport chain. The linear electron transport from water to $NADP^+$ thus requires equal efficiency of light energy conversion by the two photosystems. In changing conditions of light quality, the redox state of the interphotosystem electron carrier plastoquinone (PQ) is a key regulatory signal that governs the expression of genes encoding core protein subunits of the photosystems[7]. PQ-controlled photosystem gene expression is a regulatory component of photosystem stoichiometry adjustment, an acclimatory response that improves the quantum efficiency of photosynthesis in environments with altered light quality[8].

Chloroplast Sensor Kinase (CSK) is a modified sensor histidine kinase with extensive phylogenetic distribution in photosynthetic eukaryotes[9,10]. CSK belongs to the two-component signal transduction family of proteins, whose components are a sensor histidine kinase, phosphorylated on a histidine residue upon sensing a specific environmental cue, and a response regulator to which the phosphoryl group is transferred. Response regulators are mostly transcription factors, which bring about an appropriate physiological response through changes in gene regulation. CSK and its cyanobacterial homolog, Histidine kinase 2 (Hik2), provide the crucial signal transduction chain that links PQ with photosystem gene expression[9,11]. However, the mechanism by which these sensor kinases monitor the redox state of PQ is unclear. Here we report on the upstream signaling events in the CSK signal transduction pathway. Using purified CSK proteins from *Arabidopsis thaliana*, the diatom *Phaeodactylum tricornutum*, and the Hik2 protein from cyanobacterium *Synechocystis* sp. PCC 6803, we show all three proteins contain an evolutionarily conserved redox-responsive [3Fe-4S] cluster that is capable of sensing the redox state of the PQ pool. We further show that the redox changes in the Fe-S cluster induce conformational changes in the CSK protein and modulate its autokinase activity.

## Results

**CSK is an Fe-S protein that binds a [3Fe-4S] cluster**. In order to understand the redox sensory mechanism of CSK, we overexpressed full-length CSK proteins from *A. thaliana* (Ara-CSK) and *P. tricornutum* (Phaeo-CSK) and the full length Hik2 protein from *Synechocystis* sp. PCC 6803 (hereafter referred as Syn-CSK) in *Escherichia coli*. The CSK proteins, purified from bacteria, were dark yellow to light brown in color at a protein concentration of ~100 μM (Syn-CSK, Fig. 1a far left), a characteristic feature of proteins containing Fe-S clusters. Since Fe-S proteins purified from bacteria do not likely incorporate stoichiometric amounts of the cluster, we reconstituted CSK proteins in the presence of excess iron and sulfide. All three reconstituted proteins became dark brown in color (Fig. 1a). The CSK proteins in the oxidized state show an absorbance peak at 410 nm, typical of Fe-S proteins containing [4Fe-4S] or [3Fe-4S] clusters (Fig. 1b–d).

We further examined the presence of an Fe-S cluster in CSK by electron paramagnetic resonance (EPR) spectroscopy. At a protein concentration of 115–176 μM, the air-oxidized CSK samples show a cubane EPR signal with $g = 2.01$ (Fig. 2). This EPR signature is characteristic of a $[3Fe\text{-}4S]^{1+}$ cluster[12]. To further confirm the identity of the cluster, we recorded the temperature dependence of the EPR spectrum. The signal from the $[3Fe\text{-}4S]^{1+}$ center has a maximum intensity at 10–12 K and has a small or vanishing signal above 40 K[13]. The radical anion of sulfur has a *g*-value similar to a [3Fe-4S] cluster but displays a maximum signal intensity at 30–60 K[14]. Figure 2 shows that CSK proteins have a maximum EPR signal intensity at 12 K and a very small amplitude at 45 K for Syn-CSK. The decrease in EPR signal above 12 K is consistent with the inference that the CSK contains a cubane-like $[3Fe\text{-}4S]^{1+}$ cluster[15].

**The Fe-S cluster of CSK is redox-responsive**. The redox response of the CSK [3Fe-4S] cluster was examined using ultraviolet–visible (UV-Vis) and EPR spectroscopies. Figure 3a–c shows UV-Vis absorbance spectra of the air-oxidized and dithionite-reduced proteins. Dithionite ($E_{m7} = -660$ mV) addition resulted in reduction of the amplitude of the 410 nm peak. Further analysis of the air-oxidized and dithionite-reduced CSK proteins with EPR confirms the redox response of the [3Fe-4S] center of CSK (Fig. 3d–f). Upon reduction with 5 mM dithionite, the EPR signal was abolished, consistent with reduction of a $[3Fe\text{-}4S]^{1+}$ into an EPR silent $[3Fe\text{-}4S]^{0}$ (Fig. 3d–f). The EPR results together with the UV-Vis data (Fig. 3a–f) thus show that the [3Fe-4S] cluster of CSK is redox-responsive.

We further determined the stoichiometry of ligand-bound $Fe^{2+}$ and $S^{2-}$ in each monomer of CSK: Syn-CSK binds $3 \pm 0.07$ $Fe^{2+}$ and $4 \pm 0.85$ $S^{2-}$ and Ara-CSK binds $2.8 \pm 0.08$ $Fe^{2+}$ and $3.7 \pm 0.46$ $S^{2-}$ (Supplementary Table 1), indicating that each monomer of CSK binds one [3Fe-4S] cluster.

**The nature of the Fe-S ligands of CSK**. The sulfhydryl (-SH) group of cysteine residues typically ligates Fe ion in Fe-S clusters. Supplementary Fig. 1 shows two cysteine residues in CSK that are conserved from cyanobacteria to higher plants. The Fe-S binding by these cysteines were tested by their substitution to serine residues, followed by quantification of Fe and S ions in the reconstituted mutant proteins. The replacement of only the first conserved cysteine (C19) decreases binding of both Fe and S (Supplementary Table 1). Since coordination of a [3Fe-4S] cluster requires two more ligands in addition to C19, we further sought to identify the nature of these chemical groups. The in vitro reconstitution of CSK proteins uses dithiothreitol (DTT) as a reducing agent. Therefore, the thiol groups of DTT may act as an inadvertent ligand for Fe-S. To rule out this possibility, we removed DTT by repeated desalting of the reconstituted protein into a buffer lacking DTT using a PD10 column. To further remove any residual DTT, we performed a second purification step using ion exchange chromatography (IEC). Figure 4a shows EPR signal of the reconstituted CSK protein containing DTT, without DTT when purified by desalting, and without DTT when purified by IEC. All CSK preparations have identical EPR signals, confirming that DTT has no contribution to the ligation of the [3Fe-4S] cluster.

Further analysis of the nature of the remaining two non-cysteine ligands utilized extended X-ray absorption fine structure (EXAFS) spectroscopy. Figure 4b, c and Supplementary Figs. 2 and 3 show the X-ray absorption near-edge structure (XANES) spectra of the oxidized Syn-CSK protein. The edge position of the oxidized Syn-CSK protein is in agreement with all three Fe ions being in an $Fe^{3+}$ oxidation state (Fig. 4b). The Fourier

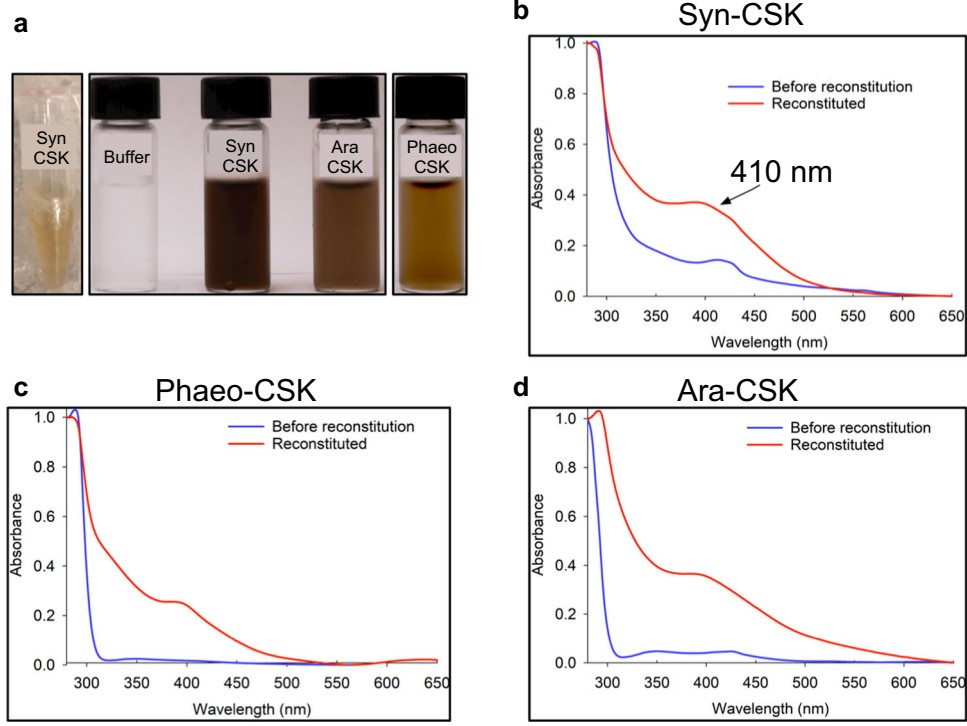

**Fig. 1 UV-Vis absorbance spectra of cyanobacterial and chloroplast CSK proteins.** Coloration of the purified and reconstituted proteins. **a** On the far-left side is Syn-CSK purified from bacteria, followed by buffer control and the reconstituted CSK proteins. UV-Vis absorbance spectra of **b** *Synechocystis* sp. PCC 6803 CSK, **c** *Phaeodactylum tricornutum* CSK, and **d** *Arabidopsis thaliana* CSK. The absorbance spectra of each CSK protein as purified from bacteria before reconstitution (blue) and after reconstitution (red) are shown.

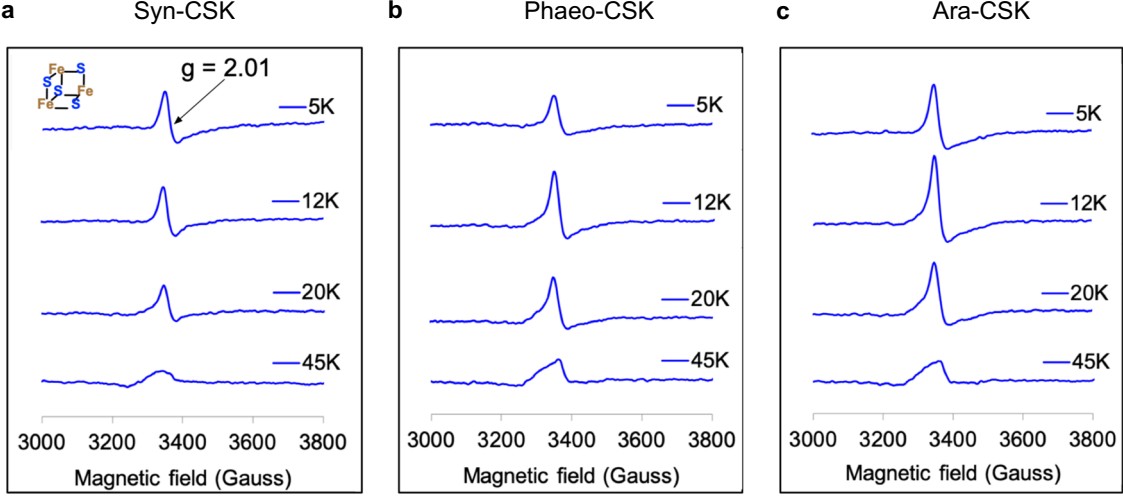

**Fig. 2 Temperature dependence of X-band EPR spectra of CSK proteins.** The X-band EPR spectrum of CSK from **a** *Synechocystis,* **b** *Phaeodactylum,* and **c** *Arabidopsis.*

transformation of the EXAFS spectrum revealed two distinct peaks (Fig. 4c). As in other Fe-S clusters, the major peak can be attributed to Fe–S interaction, while the minor peak at a larger distance can be assigned to Fe–Fe interactions[16,17]. Corresponding fits are provided in Table 1, and curves with best fits are shown in Supplementary Fig. 2. The main result from EXAFS of Syn-CSK is an Fe-S distance of ~2.18 Å, consistent with coordination of Fe by a sulfur atom in Fe–S–Fe bridges and cysteine. Addition of an Fe–O/N interaction at ~1.86–1.96 Å improves EXAFS fits, Table 1. This can be due to oxygen- and/or nitrogen-containing ligands of Fe, consistent with the other ligands being non-cysteine. Oxygen and nitrogen have similar

scattering properties and thus cannot be distinguished based on EXAFS fits alone. The best fit contains Fe–Fe bond distances of ~3.0 and ~3.4 Å.

**Redox potential of the CSK [3Fe-4S] cluster.** A potentiometric redox titration, employing EPR spin intensity as a reporter, was used to determine the redox potential of the [3Fe-4S] cluster of Syn-CSK (Fig. 5). We estimate the midpoint potential ($E_m$) of the [3Fe-4S] center of CSK to be approximately −15 mV (versus the standard hydrogen electrode) at pH 8.0. The cluster exhibits a one-electron redox behavior: $[3Fe-4S]^{1+} + 1e \Leftrightarrow [3Fe-4S]^0$ (Fig. 5).

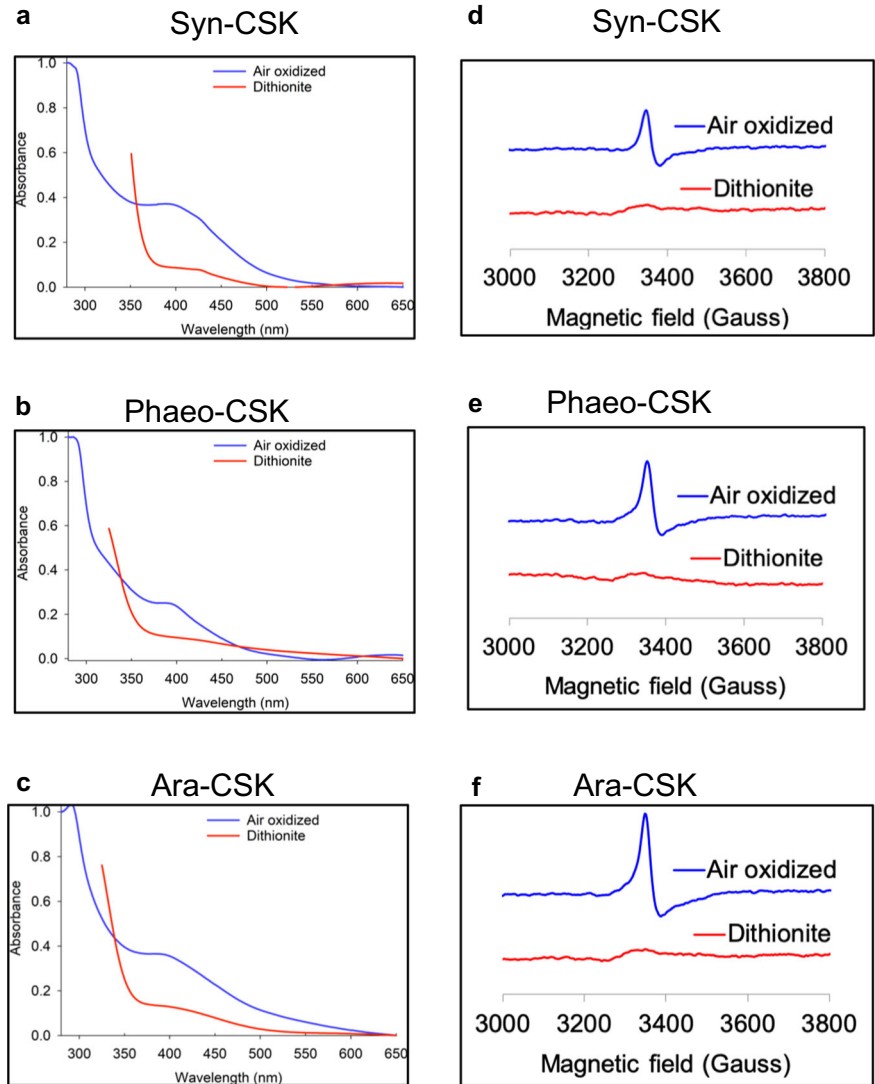

**Fig. 3 UV-Vis and EPR spectra of the oxidized and reduced CSK proteins.** The reconstituted and air-oxidized CSK protein UV-Vis spectra from Fig. 1 are replotted (blue line). The UV-Vis spectra of dithionite-reduced samples are plotted in red. UV-Vis spectra of **a** *Synechocystis* CSK, **b** *Phaeodactylum* CSK, and **c** *Arabidopsis* CSK. **d–f** EPR spectra of the air-oxidized and reduced *Synechocystis*, *phaeodactylum*, and *Arabidopsis* CSK proteins, respectively. EPR spectra were recorded at 12 K.

**Plastoquinol inhibits the kinase activity of CSK**. Having established the presence of a highly conserved redox-active [3Fe-4S] cluster within CSK, redox regulation of the autophosphorylation activity of CSK was studied. To test whether the redox state of the [3Fe-4S] cluster regulates the autokinase activity of Syn-CSK, the recombinant holoprotein was incubated with reducing or oxidizing agents in the presence of [$\gamma$-$^{32}$P]ATP. Figure 6a shows that the air-oxidized Syn-CSK is autokinase active. Treatment with reducing agents dithionite (midpoint potential at pH 7 ($E_{m7}$) of $-660$ mV), duroquinol ($E_{m7} = +7$ mV), or decylPQH$_2$ ($E_{m7} = +80$ mV) in contrast leads to the inhibition of the kinase activity of Syn-CSK (Fig. 6), indicating that reduction of the [3Fe-4S]$^{1+}$ to [3Fe-4S]$^0$ renders the kinase inactive. NADPH (Fig. 6a) or reduced ferredoxin (Fig. 6d) had no effect on the kinase activity.

The overexpressed *Arabidopsis* CSK, as purified from bacteria, is inactive as a protein kinase in vitro, possibly because of an improperly folded kinase domain or the insufficiency of the assay condition. To determine the kinase activity of plant CSK, an in vivo approach was utilized. *Arabidopsis* CSK was overexpressed in a *csk* minus mutant background. Six-week-old

*Arabidopsis* plants, containing the HA-FLAG-tagged CSK, were exposed to far-red light, an illumination condition in which CSK is active since the PQ pool exists in a relatively oxidized state. Chloroplasts were isolated from the far-red illuminated plants and the CSK protein was purified by affinity chromatography. The purified proteins were trypsin-digested and the phosphopeptides were enriched by a polymer-based metal ion affinity chromatography[18]. Liquid chromatography–mass spectrometry (LC-MS) analysis reveals that the tagged *Arabidopsis* CSK is phosphorylated on serine 188 with a phosphopeptide spectral count of two (Supplementary Table 2). A repeat of this experiment under orange light, which promotes the reduction of the PQ pool, results in no detectable phosphorylation of CSK (Supplementary Table 2).

**Fe-S cluster redox state determines CSK protein conformation**. The activity of some Fe-S proteins are regulated by their metal clusters through changes in protein oligomeric state[4]. The octameric apoSyn-CSK protein is converted into a tetramer under high concentrations of sodium ion, which is known to inhibit its autophosphorylation activity[11]. We investigated whether the

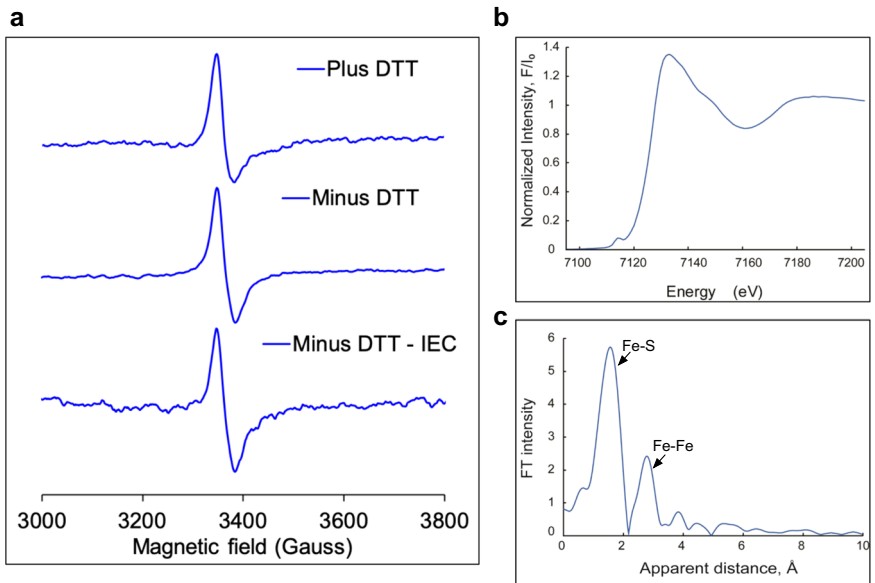

**Fig. 4 EPR and X-ray absorption spectra of Syn-CSK. a** X-band EPR spectra of desalted Syn-CSK with DTT as replotted from Fig. 2a (Plus DTT), desalted with PD10 column in a minus DTT buffer (minus DTT), and as desalted by IEC in a minus DTT buffer (IEC). **b** Normalized Fe $K$-edge XANES spectra of Syn-CSK protein. **c** Fourier-transformed EXAFS of Syn-CSK protein ($k = 3.5$–$10.0$ Å$^{-1}$).

| Table 1 Structural parameters from EXAFS fits[a] for the oxidized Syn-CSK protein. | | | | | | |
|---|---|---|---|---|---|---|
| Fit # | Shell | $R$, Å | $N^b$ | $\sigma^2$, ×10³ | $R$-factor | Reduced $\chi^2$ |
| 1 | Fe-S | 2.15 | 3 | 7.4 | 0.28 | 47,839 |
| 2 | Fe-S | 2.18 | 3 | 8.6 | 0.092 | 23,630 |
| | Fe-Fe | 3.03 | 1 | 4.5 | | |
| 3 | Fe-O | 1.91 | 1 | 1.2 | 0.03 | 16,941 |
| | Fe-S | 2.23 | 2 | 5.7 | | |
| | Fe-Fe | 3.07 | 1 | 3.1 | | |
| 4 | Fe-O | 1.96 | 2 | 3.6 | 0.017 | 8848 |
| | Fe-S | 2.26 | 2 | 10.7 | | |
| | Fe-Fe | 3.1 | 1 | 3.6 | | |
| 5 | Fe-O | 1.86 | 2 | 3.9 | 0.005 | 5092 |
| | Fe-S | 2.18 | 2 | 4.8 | | |
| | Fe-Fe | 3.03 | 1 | 6.9$^b$ | | |
| | Fe-Fe | 3.39 | 1 | 6.9$^b$ | | |

$R$-factor and reduced $\chi^2$ are the goodness-of-fit parameters (see XAS/EXAFS methods section), $S_o^2$ the amplitude reduction factor, = 1.0, was used in all fits
$R$ Fe-backscatter distance, $\sigma^2$ Debye–Waller factor
[a]Fits were done in $q$-space
[b]Denotes the value when the Debye–Waller factor ($\sigma^2$) was set to be the same for absorber and backscatter vectors

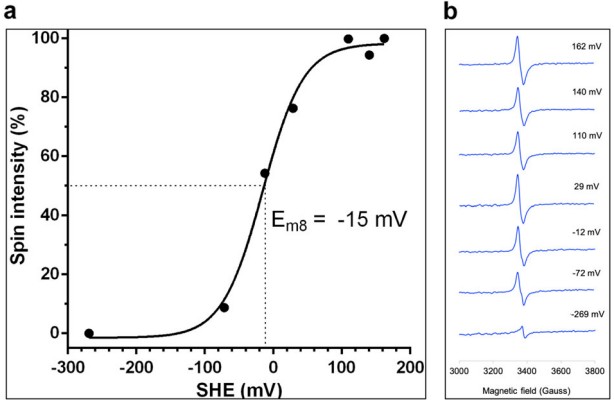

**Fig. 5 EPR reporter-based redox titration of Syn-CSK.** Starting from a redox potential of 162 mV, the sample was gradually reduced by substoichiometric amounts of dithionite in the presence of a mixture of redox mediators covering the full span of potentials shown on the x-axis. **a** The y-axis is given as a percentage of maximal spin integrations of the Fe-S cluster region. The data fit a one electron slope ($n = 1$) and give a midpoint potential $E_{m8}$ of −15 mV. The dotted line indicates 50% reduction of Syn-CSK Fe-S cluster. **b** X-band EPR spectra of Syn-CSK recorded at 12 K. Each EPR spectrum denotes samples poised at the indicated potential (in mV) after potentiometric titration of the EPR-active species.

redox changes in the [3Fe-4S] cluster control the oligomeric state of CSK by size exclusion chromatography and chemical cross-linking (Fig. 7). Figure 7a shows that both air-oxidized (blue line) and dithionite-reduced (red line) holoSyn-CSK proteins are octameric. The chemical crosslinking data (Fig. 7b) further confirm the octameric nature of CSK in both oxidizing and reducing conditions. The redox treatment thus seems not to affect CSK oligomerization. To examine whether the redox state dependence of the CSK kinase activity (Fig. 6) may be the result of an intramolecular conformational change, we analyzed the secondary structure of Syn-CSK in the presence of redox reagents by far-UV circular dichroism (CD) spectroscopy. Figure 7c, d shows the temperature dependence of melting curves of the air-exposed and potassium ferricyanide-oxidized samples. The $\alpha$-helix to random coil transition temperature of Syn-CSK measured by the amplitude of the far-UV CD spectrum at the 222 nm trough, associated principally with $\alpha$-helical structure, is higher under the air-

exposed environment (50.2 ± 1.6 °C), suggesting a more ordered and stable structure. Oxidation by potassium ferricyanide, in contrast, destabilizes or disorders the Syn-CSK to some extent, as revealed by the lower "melting" temperature (47.1 ± 0.7 °C). It is thus inferred that the oxidation of the Fe-S cluster induces an intramolecular conformational change in CSK that is conducive to its autokinase activity. Dithionite- or quinol-reduced samples could not be used in the CD analysis due to high absorbance of these redox agents in the far-UV.

**CSK is a repressor of chloroplast genes**. *Arabidopsis* plants lacking the *CSK* gene are impaired in PS I gene regulation while PS II genes are unaffected[9], consistent with an active CSK-

initiating photosystem stoichiometry adjustment through regulation of PS I amount under PQ-oxidizing far-red light condition. To examine whether the CSK-genetic control extends to non-photosystem genes, we quantified transcript levels of representative chloroplast genes in 12-day-old seedlings of wild-type Col-0, a *CSK* T-DNA knockout mutant (*csk-1*), and a *CSK*-overexpression line in the above T-DNA mutant background (CSK-OE). All genotypes were grown in white light condition. Based on our kinase assay (Fig. 6), we expect CSK to have some basal activity under white light condition where the PQ is only

partially reduced. Figure 8 shows transcript abundance of *psaA*, encoding the PsaA reaction center subunit of photosystem I; *psbA*, D1 reaction center protein of PS II; *psbD*, D2 reaction center protein of PS II; *petB*, cytochrome $b_6$ subunit of $b_6f$ complex; *ndhC*, D3 subunit of the chloroplast NAD(P)H dehydrogenase-like complex (NDH); *atpB*, β-subunit of ATP synthase; and *rbcL*, large subunit of ribulose-1,5-bisphosphate carboxylase/oxygenase (RuBisCO). The following genes were upregulated in *csk* knockout mutant line relative to wild type in a statistically significant manner (*p* value ≤ 0.05): *psaA*, *petB*, and *ndhC* (Fig. 8). Overexpression of *CSK* in a *csk* minus background rescues the mutant phenotype. The following genes were downregulated in a statistically significant manner in CSK-OE: *psaA*, *ndhC*, *atpB*, and *rbcL*, consistent with the notion that the CSK is a repressor of both photosystem and non-photosystem genes.

## Discussion

The cyanobacterial genome encodes large number of two-component systems, with as many as 146 histidine kinases in the filamentous cyanobacterium *Nostoc punctiforme* and as few as 5 histidine kinases in the small genome of the marine unicellular cyanobacterium *Prochlorococcus marinus* MED4[19]. Of the three conserved histidine kinases found in all cyanobacteria, the Hik2 (referred in this study also as Syn-CSK) is the only histidine kinase retained by chloroplasts of all major lineages of photosynthetic eukaryotes. The extensive phylogenetic distribution of Hik2 and CSK[10] suggests that these sensor kinases have an important role in both cyanobacteria and chloroplasts. CSK is

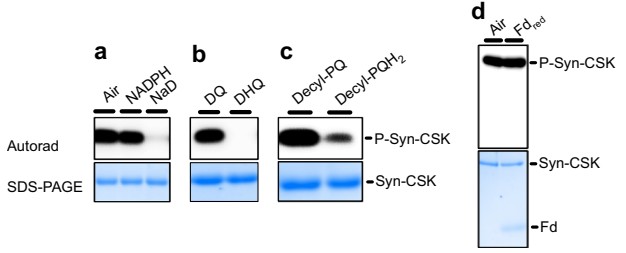

**Fig. 6 Redox control of Syn-CSK autokinase activity.** An autoradiograph showing the autokinase activity of Syn-CSK. **a** The air-oxidized CSK was pretreated with 0.25% (v/v) ethanol (control); 1 mM NADPH or 0.5 mM sodium dithionite (NaD); **b** 0.5 mM duroquinone (DQ) or 0.5 mM duroquinol (DHQ); **c** 0.5 mM decyl-plastoquinone (decyl-PQ) or 0.5 mM decyl-plastoquinol (decyl-PQH₂); **d** dithionite-reduced desalted ferredoxin. The full uncropped autoradiographs and corresponding stained gels are provided in Supplementary Fig. 8.

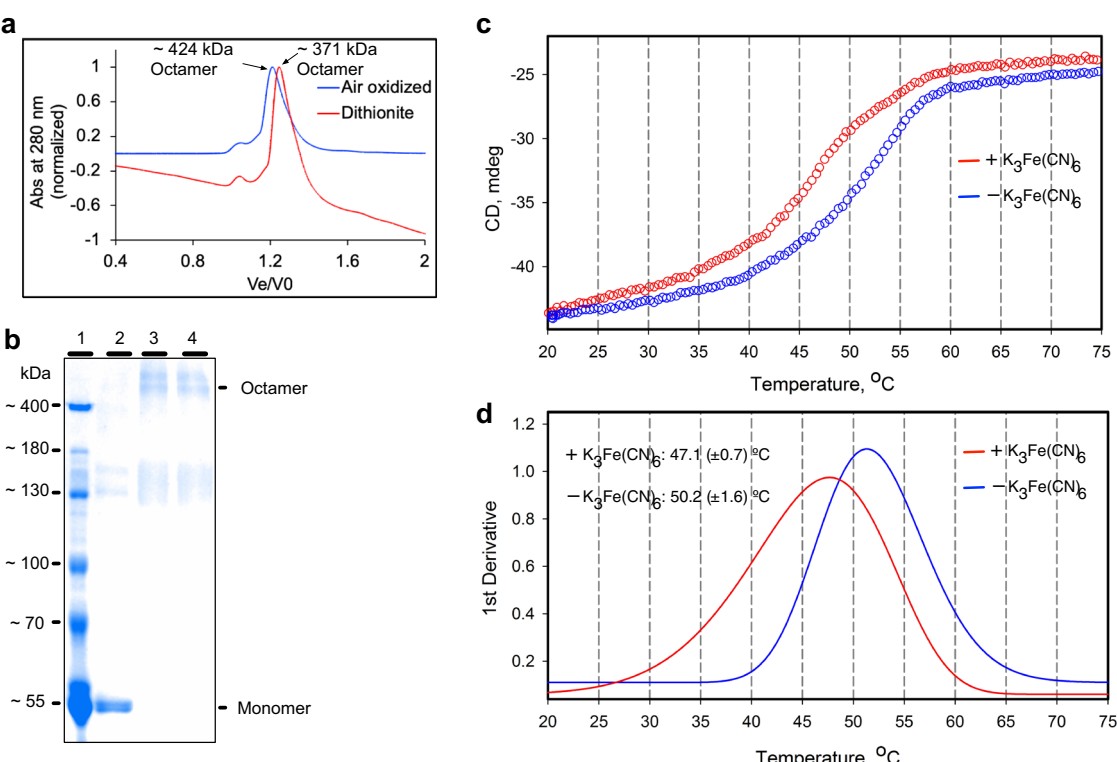

**Fig. 7 Cluster redox-state-dependent protein conformational change in CSK. a** Redox signal does not affect the oligomeric state of Hik2. The Hik2 oligomer as separated by size exclusion chromatography. Typical elution profile of the air-oxidized Hik2 protein on a Superdex 200 column as eluted with a buffer containing 20 mM Tris-HCl (pH 8) and 10 mM NaCl (blue line) or with 20 mM Tris-HCl (pH 8), 10 mM NaCl, and 2 mM sodium dithionite (red line). **b** An Urea-SDS-PAGE gel showing the products of the crosslinking reaction. Lane 1, protein molecular mass markers; lane 2, untreated Syn-CSK protein (control); lane 3, crosslinked air oxidized Syn-CSK; lane 4, crosslinked dithionite-reduced Syn-CSK. Two different oligomeric states are labeled on the right-hand side of the gel. **c** Thermal melting profiles of Syn-CSK in the absence (blue) and presence of 50 μM K₃Fe(CN)₆ (red). **d** First derivatives of melting profiles reveal the decrease in structure stability upon K₃Fe(CN)₆ oxidation of CSK [3Fe-4S] cluster (apparent Tm of 50.2 (±1.6) and 47.1 (±0.7) °C for air-exposed and K₃Fe(CN)₆-oxidized Syn-CSK, respectively).

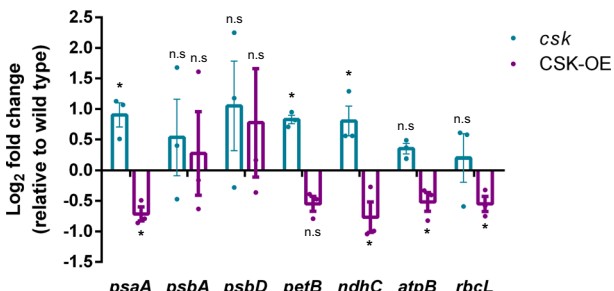

**Fig. 8 CSK is a repressor of chloroplast genes.** The Log2 fold change in the expression of selected chloroplast genes in 12-day-old *csk* mutant and CSK overexpressor (CSK-OE) are shown relative to wild type. Error bars represent ±SE from three biological replicates. For statistics, unpaired Student's *t* test was performed. *p value ≤ 0.05. n.s stands for non-significant changes.

indeed critical for linking the redox state of the PQ pool to plastid gene transcription in *Arabidopsis*[9]. The Ara-CSK protein has been shown to bind the PQ analog DBMIB (2,5-dibromo-6-iso-propyl-3-methyl-1,4-benzoquinone) in vitro with an affinity similar to known quinone-binding proteins[20], pointing to a quinone-binding pocket within CSK. However, it was not clear whether CSK senses PQ redox signal by direct binding of the quinone. In this context, we discovered a redox-responsive [3Fe-4S] cluster in cyanobacterial, diatom, and plant CSK. The molar extinction coefficient of Syn-CSK and Ara-CSK at 410 nm are 10,426 and 8,091 $M^{-1} cm^{-1}$, respectively, which is in the range expected for one [3Fe-4S] cluster per protein[21,22] and in agreement with the quantitation of the Fe-S center (Supplementary Table 1). Dithionite reduces the cluster and thus decreases the absorbance at 410 nm (Fig. 3a–c). EPR spectroscopy further confirms the redox response of the Fe-S cluster in all three CSK orthologs (Fig. 3d–f). The air-oxidized cluster is present as [3Fe-4S]$^{1+}$ with $g = 2.01$ (spin state of ½). After reduction with dithionite, the [3Fe-4S]$^{1+}$ cluster of all three CSK proteins becomes EPR-silent, consistent with a one electron reduction into a [3Fe-4S]$^{0}$ cluster with a higher spin state of 2. The UV-Vis and EPR spectroscopies thus reveal a redox-sensitive Fe-S prosthetic group in CSK, which is conserved from cyanobacteria to chloroplasts.

How is the [3Fe-4S] cluster of CSK coordinated? The Fe ion in the Fe-S cluster is typically bound by cysteines, with the thiol groups completing the tetrahedral coordination of each Fe. In some cases, aspartate or histidine residues, or backbone amides, coordinate the Fe ions, which often modify the redox potential of the cluster[23]. A few cysteine motifs are known to bind Fe-S clusters. For example, the low- and high-potential bacterial [4Fe-4S] ferredoxins contain $CX_2CX_2CX_3CP$ and $CX_2CX_{12-16}CX_{13}C$ motifs[24]. Although an obvious cysteine motif is lacking in CSK, two conserved cysteine residues are nevertheless found in its protein sequence (Supplementary Fig. 1). These spatially separated cysteine residues might be brought into proximity by protein folding such that they bind the Fe-S cluster. Cysteine-to-serine substitution of Syn-CSK, however, shows only cysteine 19 as a putative ligand of the cluster, leaving open the possibility of two non-cysteine ligands (Supplementary Table 1). Histidines, as found in Rieske Fe-S proteins, or aspartate residues, or backbone amides may fulfill this role in CSK. The EXAFS data support this possibility, revealing the presence of a short-range atomic interaction in the CSK cluster. A bond distance of around 1.86–1.96 Å corresponds to Fe–O/N interaction as arising from aspartate or histidine sidechain ligands of Fe (Table 1). The coordination of Fe with non-cysteine groups elevates the redox potential of the cluster as in the Rieske protein[23]. This, together with the O$_2$

tolerance of the CSK cluster (Supplementary Fig. 4), suggests that the cluster is likely to be a high-potential Fe-S center. The redox titration, which revealed a midpoint potential $E_{m8}$ of −15 mV (or $E_{m7}$ of +45 mV), indeed supports the high potential nature of the CSK cluster (Fig. 5). CSK contains an N-terminal GAF domain (named after *c*GMP-specific phosphodiesterases, *a*denylyl cyclases, and *F*hlA). Our observation that the conserved cysteine, involved in the coordination of the [3Fe-4S] cluster (Supplementary Table 1), resides within the GAF domain (Supplementary, Fig. 1) supports the role of this domain in redox sensing. The GAF domain may additionally be involved in binding the PQ, as shown for the *Arabidopsis* CSK, and thus bringing the PQ redox signal close to the redox-active Fe-S cluster. The Syn-CSK has further been shown to sense an additional cue—sodium ions[11]. The salt-sensing property, however, is confined to its DHp (dimerization and phosphotransfer) domain, with consequences for its oligomeric state and activity.

The PQ pool has a standard midpoint potential of +80 mV at pH 7.0 for two-electron chemistry (PQH$_2$/PQ)[25], −154 mV for one-electron chemistry (PQH$_2$/PQ$^-$ or PQ$^-$/PQ)[26], and an effective redox poise that is dependent on photosystem activity. The [3Fe-4S] cluster of CSK turns out to be an ideal cofactor for sensing PQ as both Fe-S clusters and quinones can undergo one- or two-electron redox chemistry[27,28]. Furthermore, the experimentally measured $E_{m8}$ of CSK is −15 mV (Fig. 5), which corresponds to +45 mV at pH 7.0, an $E_m$ difference of 199 mV from the one-electron reduction potential of PQ (−154 mV). The reduction of CSK Fe-S cluster by plastosemiquinol is thus a thermodynamically favorable reaction.

How does the water-soluble CSK perceive the membrane-intrinsic PQ redox signal? A few known sensors of PQ such as the STN proteins have membrane-spanning regions, which increase the likelihood of encountering the reactive phenolic head group of quinones[29]. Although sensing a membrane-intrinsic signal seems challenging for soluble proteins, some soluble sensors overcome this barrier by weakly associating with the membrane using their amphipathic helices and/or hydrophobic protein segments[30,31]. A thylakoid membrane-binding assay, utilizing a dot immunoblotting technique, finds CSK protein to be specifically interacting with the thylakoid (Supplementary Fig. 5a). Helical wheel modeling of the GAF domain of CSK indeed reveals a conserved amphipathic helix (Supplementary Fig. 5b, c), which may facilitate transient association with the surface of the thylakoid membrane. Alternatively, the amphipathic helix of CSK may afford membrane access through association with a membrane protein. In this regard, it is interesting to note that the Hik2 protein interacts with the phycobilisome linker protein ApcE in a yeast two-hybrid assay[32]. The observation that the electron transport from quinones can occur over a distance of 10–20 Å[33], half of the thylakoid membrane span, further supports the suggested in vivo CSK [3Fe-4S] cluster reduction by PQH$_2$.

The redox-responsive autophosphorylation and the transphosphorylation of the response regulator are critical to CSK's function of transducing PQ redox signals to photosystem gene expression machinery during the adjustment of photosystem stoichiometry. A direct demonstration of the redox control of CSK kinase activity has, however, not been provided previously. Evidence for the redox regulation of the Syn-CSK is provided in this study (Fig. 6). Syn-CSK autokinase activity is higher under oxidized conditions (Fig. 6), consistent with a predominantly active CSK in the presence of an oxidized PQ pool. Treatment with reducing agents: dithionite, duroquinol, or decylPQH$_2$ inhibit Syn-CSK (Fig. 6a–c). The inhibition by decylPQH$_2$ is especially relevant as it points to an in vivo suppression of CSK activity by PQH$_2$. Though spinach Fd ($E_{m7}$ = −420 mV) is thermodynamically capable of reducing the Fe-S cluster of CSK,

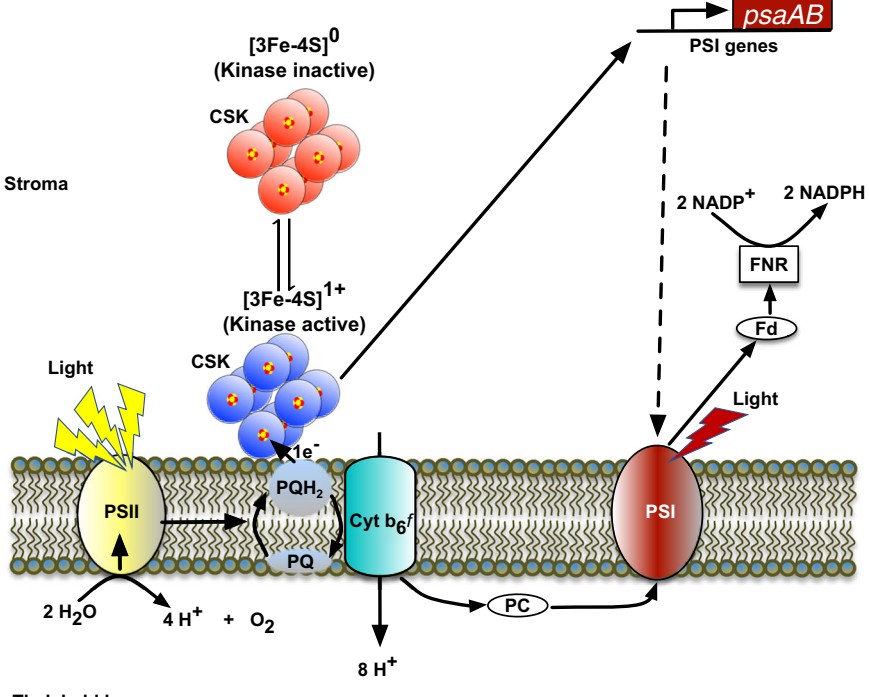

**Fig. 9 Hypothetical model of CSK as a PQ redox sensor.** The yellow light, which preferentially excites PS II, leads to the reduction of the PQ pool. The PQH$_2$ reduces the CSK Fe-S cluster, rendering CSK into an inactive kinase. The far-red light under shaded conditions leads to the preferential excitation of PSI and in turn to the oxidation of the PQ pool. Under this light condition, the Fe-S cluster of CSK is reoxidized by oxygen, converting CSK into an active kinase from the inactive reduced conformation. The active CSK regulates photosystem gene expression. The CSK protein is shown as an octamer.

we find no evidence for this reduction in our kinase assay (Fig. 6d). Since redox reactions with Fd require a specific protein–protein interaction surface and SH3-like domains[34], it is likely that PQH$_2$ is the only physiologically relevant in vivo reductant of CSK. The possession of an Fe-S cluster seems to suppress the autokinase activity of CSK as the apoSyn-CSK is catalytically more active than the holoenzyme (Supplementary Fig. 6). Furthermore, the apoprotein does not respond to redox treatment[11]. The incorporation of an Fe-S cluster thus renders CSK amenable to redox regulation. An analogy for the role of the Fe-S cluster is that it acts as a break for CSK kinase activity, with the reduction of the cluster equivalent to fully depressing the break pedal and the oxidation releasing it.

In cyanobacteria, the CSK homolog, Hik2, interacts with two response regulators Rre1 and RppA[11,35], both of which seem to be crucial for the regulation of photosystem genes by a mechanism dependent on the PQ redox state[35]. Rre1 is found in certain non-green algae such as diatoms as either a chloroplast or a nuclear gene product, with the cognate sensor kinase CSK occurring as a canonical histidine kinase[10]. Neither Rre1 nor RppA are found in green algal and plant chloroplasts. The lack of a typical response regulator partner seems to correlate with the loss of CSK's conserved histidine autophosphorylation site. In the green lineage, CSK has been suggested to work instead as a serine/threonine kinase, phosphorylating the sigma factor 1 (SIG1) transcription initiation factor subunit of the chloroplast eubacterial RNA polymerase[20]. Although biochemical evidence for SIG1 phosphorylation by CSK is lacking, our demonstration of serine 188 autophosphorylation in *Arabidopsis* CSK (Supplementary Table 2) is consistent with that proposal. The plant CSK thus seems to have acquired a serine/threonine-type catalytic mechanism similar to plant phytochromes[36], ethylene receptors[37], and pyruvate dehydrogenase kinase[38]—serine/threonine kinases, which share histidine kinase ancestry.

Based on the work presented here and elsewhere[9,11,20,39], we propose that under light utilized mainly by photosystem II, which promotes the reduction of the PQ-pool, the [3Fe-4S]$^{1+}$ of CSK will be reduced to [3Fe-4S]$^0$ by the PQ pool, which in turn inactivates CSK through cluster redox-induced protein conformational changes (Figs. 7 and 9). In the absence of a sustained PQH$_2$ inhibitory signal, the oxygen oxidizes the Fe-S cluster, in turn activating CSK. In plant chloroplasts, the active CSK functions as a sigma factor kinase[17,36] to inhibit transcription of specific chloroplast genes (Fig. 8). In cyanobacteria and non-green algae, where the CSK is present with its cognate response regulator partner(s), the active CSK undergoes autophosphorylation on its conserved histidine residue and transfers a phosphoryl group to Rre1 and RppA. Phospho-Rre1 and phospho-RppA then regulate photosystem and antenna genes. While the specific downstream gene regulatory mechanisms leading to stoichiometry adjustment of photosynthetic electron transport complexes in cyanobacteria and plants remain to be determined, the present study contributes to the elucidation of the key upstream signaling events in this crucial light-acclimatory pathway of oxygenic photosynthesis.

## Methods

**Construction of recombinant plasmids**. Coding sequences of full-length *Synechocystis* sp. PCC6803 Hik2 (*slr1147*), *A. thaliana* CSK (*At1g67840*), and *P. tricornutum* CSK (*PHATRDRAFT_41268*) were amplified by using the primer pairs listed in Supplementary Table 3. PCR products of Hik2 and *Arabidopsis* CSK were digested with NdeI and XhoI endonucleases (New England BioLabs) and cloned into pET-21b (Novagen) expression vector digested with the same enzymes. The *Phaeodactylum* CSK PCR product was digested with KpnI and XhoI and cloned into pETG-41A (EMBL) expression vector. The identities of the recombinant clones were confirmed by sequencing.

**Site-directed mutagenesis of conserved cysteines**. Mutagenesis of the conserved cysteine residues of the *Synechocystis* Hik2 (C19, C35, and C153) to serines

were made using the Stratagene QuickChange Site-directed Mutagenesis Kit. The primer pairs used are listed in Supplementary Table 3. Mutagenesis was confirmed by sequencing.

**Expression and purification of recombinant CSK**. The BL21 (DE3) Rosetta chemically competent cells (Stratagene) were transformed with CSK expression constructs. Transformed bacterial colonies, grown on agar plates, were used to inoculate starter cultures (10 mL each) in Luria Broth (LB) growth media with 100 µg mL$^{-1}$ ampicillin. Each culture was grown overnight, then diluted 1:100 in 1 L LB media and grown at 37 °C to an optical density at 600 nm of ~0.55, before inducing protein expression with 0.5 mM IPTG (Fisher Scientific). Bacterial cultures were grown for an additional 16 h at 16 °C. Cells were harvested by centrifugation at 6000 rpm for 10 min at 4 °C, and the pellet re-suspended in buffer containing 300 mM NaCl, 20 mM Tris-HCl, pH 8.0, 25 mM imidazole, 5 mM DTT, and 1 mM PMSF. The cells were lysed with an EmulsiFlex-C3 homogenizer (Avestin). The lysate was clarified by centrifugation at 35,000 × g for 20 min at 4 °C. The supernatant was applied to a HisTrap$^{TM}$ HP 5 mL Ni$^{2+}$-charged affinity chromatography column (GE Healthcare Life Sciences), and the recombinant protein was purified according to the manufacturer's instructions. The purified proteins were buffer exchanged into 20 mM Tris-HCl, pH 8, and 100 mM NaCl before further analysis.

**In vitro reconstitution of the Fe-S cluster**. The reconstitution procedure was carried out in an anaerobic glove bag (Glas-Col) filled with argon gas. The desalted Syn-CSK, Ara-CSK, or Phaeo-CSK proteins at a concentration of 8.4 mg mL$^{-1}$ were reduced with 10 mM DTT anaerobically and incubated for 30 min on ice. After this pretreatment, equal amounts of ferrous ammonium sulfate and sodium sulfide were slowly added at a stoichiometry of 15:1 (iron or sulfide to protein) and incubated at 22 °C for 3.5 h. Precipitant was removed by centrifugation at 5000 × g at 4 °C for 5 min and the unbound iron and sulfide removed by buffer exchange with 2 mM DTT, 100 mM NaCl, and 20 mM Tris-HCl, pH 8.0 using a PD-10 column.

**Acid-labile sulfide and iron quantification**. Acid-labile sulfide content was measured as in ref. [40] by formation of methylene blue (absorbance maximum at 666 nm) from the reaction of N,N-dimethyl-p-phenylenediamine with H$_2$S and excess FeCl$_3$. The same method was downscaled 12-fold for determination of sulfide in 96-well plates (200 µL sample volume) and confirmation by visible spectroscopy that methylene blue was produced. Standardization was carried out with freshly purchased Na$_2$S. This method is highly specific for acid-labile sulfide and does not give a response with commonly encountered sulfur compounds, including protein-bound cysteine or methionine[41]. Iron was quantified by using the commercial Quantichrom Iron Assay Kit purchased from Bioassay Systems. Protein concentrations for assays and Fe/S analysis were determined by the Bradford method, using bovine serum albumin as a standard.

**UV-Vis and EPR spectroscopy**. UV-Vis absorbance spectra were recorded at room temperature with a HITACHI U-3900H spectrophotometer. The EPR spectrum was recorded with a Bruker ELEXSYS E580 spectrometer, operating at X-band, and the temperature controlled using an Oxford ITC503 cryostat system. Experiments were set to achieve maximum signal-to-noise ratio without producing saturation and/or distortion of EPR signals. The EPR parameters were: microwave power, 10 mW; modulation amplitude, 10 G; modulation frequency, 100 kHz at 5–45 K. Reduced and oxidized samples of the protein (Syn-CSK 173 µM, Phaeo-CSK 115 µM, and Ara-CSK 176 µM) were prepared in a buffer containing 2 mM DTT, 100 mM NaCl, and 20 mM Tris-HCl, pH 8.0. The oxidized protein was obtained by incubation in air at 22 °C for 5 min and reduced protein by addition of 5 mM dithionite dissolved in 1 M Tris-HCl, pH 8, followed by incubation at 22 °C for 5 min. Two hundred microliters of oxidized or reduced samples, taken in quartz tubes, were immediately flash frozen in liquid nitrogen for EPR analysis.

**Anion exchange chromatography**. The reconstituted Syn-CSK was first desalted into 10 mM NaCl and 25 mM HEPES, pH 8 with a PD10 column, before being subjected to DEAE-Sepharose CL6B IEC (Sigma-Aldrich). The ion exchange column was washed with 2 column volume of equilibration buffer and the protein was eluted with 5 column volume 300 mM NaCl and 25 mM HEPES, pH 8. Two hundred microliters of the concentrated eluate was taken in quartz tubes and immediately flash frozen in liquid nitrogen for EPR analysis.

**X-ray absorption spectra (XAS) and EXAFS measurements**. XAS were collected at the Advanced Photon Source (APS) at Argonne National Laboratory on bending magnet beamline 20 having electron energy 7 GeV and average current of 100 mA. The radiation was monochromatized by a Si(110) crystal monochromator. The intensity of the X-rays was monitored by three ion chambers ($I_0$, $I_1$, and $I_2$) filled with 30% nitrogen and 70% helium and placed before the sample ($I_0$) and after the sample ($I_1$ and $I_2$). Iron foil was placed between $I_1$ and $I_2$ and its absorption was recorded with each scan for energy calibration. In all, 1.2 mmol of Syn-CSK protein was prepared in a reaction buffer containing 0.1 M NaCl and 0.02 M Tris-HCl (pH

8.0). Plastic (Lexan) EXAFS sample holders (inner dimensions of 12 mm × 2 mm × 3 mm) filled with frozen samples were inserted into a cryostat pre-cooled to 20 K. The samples were kept at 20 K in a He atmosphere at ambient pressure. Data were recorded as fluorescence excitation spectra using a 13-element energy-resolving detector. In order to reduce the risk of sample damage by X-ray, defocused mode (beam size 1 × 1.6 mm) was used and no damage was observed. The shutter was synchronized with the "scan software" preventing exposure to X-rays in between scans. Fe XAS energy was calibrated by the maximum of the pre-edge feature of the iron foil (7112 eV), which was placed between $I_1$ and $I_2$ ionization chambers. EXAFS scans with 5 eV steps in the pre-edge region, 0.5 eV steps through the edge, and 0.05 Å$^{-1}$ steps from $k = 2.0$ to 14 Å$^{-1}$ were used.

Athena software was used for data processing[42]. The energy scale for each scan was normalized using Fe foil standard and multiple scans for the same samples were added together. The data in energy space were pre-edge corrected, normalized, and background corrected. The processed data were then converted to the photoelectron wave vector ($k$) space and weighted by $k^3$. The electron wave number is defined as $k = [2m(E - E_0)/\hbar^2]^{1/2}$, where $E_0$ is the energy origin or the threshold energy. K-space data were truncated near the zero crossings ($k = 4.0–10.0$ Å$^{-1}$) before the Fourier transformation in R-space. The $k$-space data were transferred into the Artemis Software for curve fitting. In order to fit the data, the Fourier peaks were isolated by applying a Hanning window to the first and last 15% of the chosen range, leaving the middle 70% untouched. Curve fitting was performed using ab initio-calculated phases and amplitudes from the FEFF9 program from the University of Washington. Ab initio-calculated phases and amplitudes were used in the EXAFS equation[43]:

$$\chi(k) = S_0^2 \sum_j \frac{N_j}{kR_j^2} f_{eff_j}(\pi, k, R_j) e^{-2\sigma_j^2 k^2} e^{\frac{-2R_j}{\lambda_j(k)}} \sin(2kR_j + \emptyset_{ij}(k)) \qquad (1)$$

where $N_j$ is the number of atoms in the $j$th shell; $R_j$ the mean distance between the absorbing atom and the atoms in the $j$th shell; $f_{eff_j}(\pi, k, R_j)$ is the ab initio amplitude function for shell $j$, and the Debye–Waller term $e^{-2\sigma_j^2 k^2}$ accounts for damping due to static and thermal disorder in absorber–backscatterer distances. The mean free path term $e^{\frac{-2R_j}{\lambda_j(k)}}$ reflects losses due to inelastic scattering, where $\lambda_j(k)$ is the electron mean free path. The oscillations in the EXAFS spectrum are reflected in the sinusoidal term $\sin(2kR_j + \emptyset_{ij}(k))$, where $\emptyset_{ij}(k)$ is the ab initio phase function for shell $j$. This sinusoidal term shows the direct relation between the frequency of the EXAFS oscillations in $k$-space and the absorber–backscatterer distance. $S_0^2$ is an amplitude reduction factor representing central atom shake-up and shake-off effects. The mean free path of the electron ($\lambda$) is due to the finite core hole lifetime and interactions with the valence electrons.

The EXAFS equation (Eq. 1) was used to fit the experimental data using $N$, $S_o^2$, $E_0$, $R$, and $\sigma^2$ as variable parameters (see fit results in Table 1). $N$ refers to the number of coordination atoms surrounding Fe in each shell. The quality of fit was evaluated by the $R$-factor and the reduced $\chi^2$ value. $R$-factor is used to denote closeness of fit. An $R$-factor <2% denotes that the fit is good enough, whereas $R$-factor between 2% and 5% denotes that the fit is correct within a consistently broad model[44]. The reduced $\chi^2$ value is used to compare fits as more absorber–backscatter shells are included to fit the data. Reduced $\chi^2$ value justifies the inclusion of an additional shell.

**Potentiometric redox titration**. The protocol followed was modified from refs. [45,46]. Five milliliters of 8.4 mg mL$^{-1}$ Syn-CSK protein was prepared in a reaction buffer containing 0.1 M NaCl, 0.05 M Tris-HCl (pH 8.0), and 10% glycerol. Forty micromolar of each of the following mediators: 1,4-benzoquinone ($E_{m7} = +280$ mV), 1,2-naphthoquinone ($E_{m7} = +135$ mV), 1,4-naphthoquinone ($E_{m7} = +65$ mV), duroquinone ($E_{m7} = +7$ mV), 2,5-dihydroxy-1,4-benzoquinone ($E_{m7} = -60$ mV), anthraquinone-2,6-disulfonic acid disodium salt ($E_{m7} = -184$ mV), and sodium anthraquinone-2-sulfonate ($E_{m7} = -225$ mV), were added to the protein solution and allowed to equilibrate for 30 min before recording potential. The potential was gradually lowered by adding substochiometric amount of dithionite dissolved in argon-bubbled 1 M Tris-HCl (pH 8). Two hundred microliters of CSK protein sample, poised at a desired potential, was withdrawn with syringe and was injected into an EPR tube and frozen in liquid nitrogen for EPR analysis. Anaerobic conditions were maintained by flushing the cuvette with water-saturated argon. The redox electrode system was calibrated with saturated quinhydrone dissolved in 0.1 M sodium phosphate, pH 7. The accuracy of the $E_m$ measurement was checked at the negative potential by titration of 20 µM FMN, which has a midpoint potential of −205 mV at pH 7.0. The midpoint potential was calculated using a derivation of the Nernst equation:

$$EPR\ signal = \frac{Maximum\ EPR\ signal}{1 + e^{(E_m - E)\frac{F}{RT}}}$$

where $E$ = potential in voltage,
$E_m$ = midpoint potential in voltage,
$F$ = Faraday constant = 96,480 C mol$^{-1}$,
$R$ = Gas constant = 8.314 J K$^{-1}$ mol$^{-1}$, and
$T$ = Temperature in Kelvin.

**In vitro autophosphorylation**. Autophosphorylation was carried out in a final reaction volume of 25 μL. The reaction contained 20 μL of 2 μM reconstituted Syn-CSK, which was desalted into 1.25-fold concentrated kinase reaction buffer containing 62.5 mM Tris-HCl at pH 8, 6.25 mM KCl, 12.5% glycerol, and 6.25 mM MgCl$_2$. Where indicated, 1 μL redox reagent was added to the protein, with the specified final concentrations: 1 mM NADPH (dissolved in water), 0.5 mM sodium dithionite (dissolved in 1 M Tris-HCl, pH 8), 0.5 mM duroquinone (dissolved in 95% ethanol), 0.5 mM duroquinol (dissolved in 95% ethanol), 0.5 mM decyl-PQ (dissolved in 95% ethanol), and 0.5 mM decyl-plastoquinol (dissolved in 95% ethanol). The reduced quinones were prepared by bubbling oxidized quinones with hydrogen gas in the presence of a platinum catalyst. Spinach ferredoxin (Sigma-Aldrich) was first reduced with 2 mM dithionite in a glove bag for 5 min and desalted into kinase reaction buffer lacking dithionite. Six micrograms of the reduced and desalted ferredoxin was then added to Syn-CSK. The kinase reaction tube was covered with 5 μL of mineral oil in order to keep the gas phase at a minimum and incubated at 22 °C for 5 min. The autophosphorylation reaction was initiated by the addition of 4 μL of a 6.25-fold concentrated ATP solution containing 2.5 mM disodium ATP (Sigma) and 2.5 μCi [γ-$^{32}$P]ATP (6000 Ci mmol$^{-1}$) (PerkinElmer). Reactions were incubated for 60 s at 22 °C, and the autophosphorylation reaction terminated by the addition of 6.25 μL of 5-fold concentrated Laemmli sample buffer[47]. Reaction products were separated by sodium dodecyl sulfate–6 M urea–11% (w/v) polyacrylamide gel electrophoresis. The gel was rinsed with gel running buffer and subsequently exposed to a phosphor plate overnight. The incorporated γ-$^{32}$P was visualized by autoradiography.

**Immunopurification and phosphopepide identification of CSK**. Six-week-old *Arabidopsis* plants, containing the HA-FLAG-tagged CSK, were transferred to far-red light. Chloroplasts were isolated from the illuminated plants and the CSK protein was purified by the FLAG-HA Tandem Affinity Purification Kit (Sigma-Aldrich). The purified proteins were subjected to trypsin digestion, phosphopeptide enrichment[18], and LC-MS/MS analysis.

**Size exclusion chromatography**. The oligomeric state of Syn-CSK was determined by subjecting the purified protein to Superdex 200 10/300GL Increase (GE Healthcare Life Sciences) size exclusion chromatography, equilibrated with 20 mM Tris-HCl (pH 8.0) and 20 mM NaCl. The molecular mass of CSK was determined by using the calibration curve at 20 mM NaCl and 20 mM Tris-HCl (pH 8.0) (Supplementary Fig. 7).

**Chemical crosslinking**. Syn-CSK protein was desalted into crosslinking reaction buffer (25 mM HEPES-NaOH at pH 8.0, 5 mM KCl, and 5 mM MgCl$_2$) using a PD-10 desalting column. Chemical crosslinking was carried out in a total reaction volume of 1.0 mL containing 4 μM of Syn-CSK protein. The Syn-CSK was treated with 0.5 mM dithionite under anaerobic condition or air oxidized for 5 min. The crosslinking agent disuccinimidyl suberate (DSS) was added from a 24.7 mM stock solution in dimethyl sulfoxide to give a final DSS concentration of 2 mM. Reactions were incubated at 22 °C for 30 min. Reactions were stopped by addition of a solution containing 50 mM Tris-HCl and 10 mM glycine, pH 7.5.

**Far-UV CD analysis of thermal stability**. Far-UV CD spectra and structure stability were measured in stirred quartz cuvettes with an optical path length of 1 cm using a Chirascan spectropolarimeter (Applied Photophysics) in 20 mM Tris, pH 7.8, and 0.1 NaCl. Thermal denaturation profiles were obtained by measuring the amplitude of the CD signal at 222 nm over a temperature range of 20–80 °C with a rate of temperature increase of 1 °C min$^{-1}$ and step of 0.4 °C. Changes in protein thermal stability caused by oxidation by 50 μM K$_3$Fe(CN)$_6$ were detected by a comparison of peak positions of the first derivative of the CD amplitude versus thermal melting profiles.

**Quantitative real-time PCR**. Total RNA was isolated from the leaves of 12-day-old seedlings using the ZR Plant RNA MiniPrep Kit (Zymo Research). RNA was treated with RNase-free DNase (Zymo Research) to eliminate possible DNA contamination. First-strand cDNA was synthesized from 0.400 μg of RNA with the RevertAid First Strand cDNA Synthesis Kit (Fisher Scientific). Real-time quantitative reverse transcriptase polymerase chain reaction was performed with a one-step QuantiNova SRBR Green PCR Kit (Qiagen) in a StepOnePlus thermocycler (Applied Biosystems). The expression values of target genes were normalized to both total RNA and endogenous *Actin8* control. The relative changes in gene expression were analyzed by $2^{-\Delta\Delta Ct}$ method.

**Statistics and reproducibility**. Statistical significance of data were tested by unpaired Student's *t* test. The sample size (*n*) and the nature of replicates have been given wherever relevant.

**Reporting summary**. Further information on research design is available in the Nature Research Reporting Summary linked to this article.

## Data availability

The dataset generated and analyzed in the current study is available as Supplementary Data 1. All other data (if any) are available upon request.

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

## Acknowledgements

We thank the Purdue University Biochemistry Department and Showalter Trust for funding and Dr. Robert M. Everly and James R. Zimmerman for assistance with EPR spectroscopy. W.A.C. and S.D.Z. are supported by DOE DE-SC0018238. Y.P. was supported by NSF, CHE-1900476. The use of the Advanced Photon Source, an Office of Science User Facility operated by the U.S. Department of Energy (DOE) Office of Science by Argonne National Laboratory, was supported by the U.S. DOE under Contract DE-AC02-06CH11357. The PNC/XSD (Sector 20) facilities at the Advanced Photon Source and research at these facilities were supported by the U.S. Department of Energy, Basic Energy Science and the Canadian Light Source. Publication of this article was funded in part by Purdue University Libraries Open Access Publishing Fund.

## Author contributions

I.M.I. and S.P. designed experiments and analyzed data with assistance from W.A.C., Y.P. and S.D.Z.; I.M.I., H.W., G.E.K., S.D.Z., Y.P., R.E. and Y.D. conducted experiments; I.M.I. and S.P. wrote the manuscript with inputs from W.A.C., Y.P., W.A.T. and S.D.Z.

## Competing interests

The authors declare no competing interests.
