## [Peer Review File · Communications Biology]

Reviewers' comments:

Reviewer #1 (Remarks to the Author):

This is an excellent and authoritative paper reporting the presence, activity and physiological function of a redox sensitive FeS cluster in the chloroplast sensor kinase CSK. Ibrahim et al. show that CSK has one cluster per monomer, they determine its midpoint potential, one electron activation/inactivation and they present a coherent model for its regulation that is well supported in data. This kinase, conserved throughout cyanobacteria, algae and higher plants, transmits information about the redox state of the PQ pool to the chloroplast transcriptional apparatus. The paper is extremely well-prepared and data rich. My only suggestion for improvement, beyond edits listed below, would be to change the proposed structure of CSK in the final figure to a more symmetric (cubic) arrangement of 8 monomers. Otherwise the paper was a pleasure to read. It represents a substantial step forward in understanding light mediated cyanobacterial and chloroplast gene expression regulation. As such it will be of great interest to the photosynthesis community in general. In my view it should be published with high priority.

line 200: a cluster per protein

—>one cluster per protein

Line 224: define Em8

line 225 classifies CSK cluster to be a high potential one (Fig. 5)

—> indicates the presence of a high potential cluster in CSK

line 225: explain acronym GAF at first use

line 284: indicates that this is a tenable inference

—> is consistent with that proposal

line 299: unraveled

—> determined

w. martin

Reviewer #2 (Remarks to the Author):

The paper entitled « An evolutionarily conserved iron-sulfur cluster underlies redox sensory function of the Chloroplast Sensor Kinase » by Ibrahim and collaborators describes the in vitro spectroscopic characterization of a bacterial-type two-component sensor kinase, named CSK, showing that recombinant proteins from three organisms i.e., *Arabidopsis thaliana*, *Phaedodactylum* and *Synechocystis*, bind a high potential redox-responsive [3Fe-4S] center. It is also demonstrated that reduced plastoquinone suppresses its autokinase activity and that redox changes of the Fe-S cluster translate into conformational changes. All these data suggest that this is how CSK perceives signals from the electron transport chain to link its status to the chloroplast gene expression.

Overall, the paper reports on very interesting observations which remain, however, at the in vitro level. Obtaining evidence for the presence of an Fe-S cluster in CSK in vivo or for changes in the redox state is somehow mandatory to validate the findings.

I have listed below several other concerns that the authors might consider to make the story more convincing/appealing.

1. I understand from the text (line 86) that full-length proteins from *A. thaliana* and *Phaeodactylum* have been expressed in *E. coli*. Does it mean that the chloroplastic targeting signal has not been removed? This would be a problem as the expressed proteins would not correspond to their physiological mature forms. This may explain why the recombinant Arabidopsis CSK has no kinase activity (lines 150-151).

2. From the mutagenesis results, it is not clear what are the ligands of the Fe-S cluster. In fact, it is puzzling that the only cysteine, which seems to contribute to the binding (Cys19), is not conserved in orthologs from photosynthetic organisms (see Fig S1).

This brought me several comments :

a. In vitro Fe-S cluster reconstitution may result in artifacts when the reductant (DTT here) is not removed as the thiol groups may serve as ligands. Second, PD10 is often not enough to remove reconstitution reagents and this likely requires an ion exchange chromatography step. Third, in line 338, what means precipitant?

b. Further spectroscopic analysis (resonance Raman for instance) are required to identify the nature of the ligands.

3. The observation that residues possibly forming an amphipathic helix are present in the GAF domain of CSK is interesting but the authors have to follow up this idea and demonstrate whether or not it can promote attachment to thylakoid membranes.

4. It is not clear how redox changes on the Fe-S cluster in CSK can translate into changes in gene expression (as proposed in the model) as the kinase activity of CSK seems independent on the presence of the Fe-S cluster (supplementary fig 3).

5. With the phosphopeptide enrichment method, do we get information on the level of protein phosphorylation? This would be nice to have such an estimate and correlation with the status of the electron transport chain.

By the way, in line 357 : give details or reference again (as in line 158) on how phosphopeptide enrichment was performed.

Minor comments :

Line 172 : redox reagents

Line 330 NaCl

Line 353 : phosphopeptide

Line 725 : I would suggest to remove « by air ».

Reviewer #3 (Remarks to the Author):

The chloroplast protein kinase CSK was identified more than 10 years ago. Whether this kinase plays an important role in adjusting chloroplast gene expression in response to changes in the redox state of the photosynthetic electron transport chain remains a controversial issue and its main role is still unclear. In this manuscript the authors have performed an extensive and thorough biochemical analysis of CSK from different photosynthetic organisms and they show that it contains a 3Fe-4S center and that its autocatalytic kinase activity is activated under oxidizing conditions and inactivated

under reducing conditions. This is clearly an important finding but this study does not provide further clues on the functional role of this kinase.

Additional comments

Statements such as "PQ-controlled photosystem gene expression forms the basis of the photosystem stoichiometry adjustment" (l. 63-64) are controversial as most of the current literature does not support such a claim.

Table 2 lacks a control of the kinase activity under reducing conditions

Reviewers' comments:

Reviewer #1 (Remarks to the Author):

This is an excellent and authoritative paper reporting the presence, activity and physiological function of a redox sensitive FeS cluster in the chloroplast sensor kinase CSK. Ibrahim et al. show that CSK has one cluster per monomer, they determine its midpoint potential, one electron activation/inactivation and they present a coherent model for its regulation that is well supported in data. This kinase, conserved throughout cyanobacteria, algae and higher plants, transmits information about the redox state of the PQ pool to the chloroplast transcriptional apparatus. The paper is extremely well-prepared and data rich. My only suggestion for improvement, beyond edits listed below, would be to change the proposed structure of CSK in the final figure to a more symmetric (cubic) arrangement of 8 monomers. Otherwise the paper was a pleasure to read. It represents a substantial step forward in understanding light mediated cyanobacterial and chloroplast gene expression regulation. As such it will be of great interest to the photosynthesis community in general. In my view it should be published with high priority.

We thank reviewer 1 for the positive evaluation of our manuscript. The proposed structure of CSK has now been modified to accommodate the reviewer's suggestion. Each CSK monomer is now represented by a circle, which together form a cubic octamer (Fig. 10, revised manuscript).

line 200: a cluster per protein
—>one cluster per protein

Done. Line 253, revised manuscript.

Line 224: define E_{m8}

Done. Line 279, revised manuscript.

line 225 classifies CSK cluster to be a high potential one (Fig. 5)
—> indicates the presence of a high potential cluster in CSK

This sentence now reads: “The redox titration, which revealed a midpoint potential E_{m8} of -15 mV (or E_{m7} of +45 mV), indeed supports the high potential nature of the CSK cluster (Fig. 6).

line 225: explain acronym GAF at first use

Yes, done. “...CSK contains an N-terminal GAF domain (named after cGMP-specific phosphodiesterases, adenylyl cyclases and *FhlA*)...”

line 284: indicates that this is a tenable inference
—> is consistent with that proposal

Done. Line 341, revised manuscript.

line 299: unraveled

—> determined

Done.

w. martin

Reviewer #2 (Remarks to the Author):

The paper entitled « An evolutionarily conserved iron-sulfur cluster underlies redox sensory function of the Chloroplast Sensor Kinase » by Ibrahim and collaborators describes the in vitro spectroscopic characterization of a bacterial-type two-component sensor kinase, named CSK, showing that recombinant proteins from three organisms i.e., *Arabidopsis thaliana*, *Phaeodactylum* and *Synechocystis*, bind a high potential redox-responsive [3Fe-4S] center. It is also demonstrated that reduced plastoquinone suppresses its autokinase activity and that redox changes of the Fe-S cluster translate into conformational changes. All these data suggest that this is how CSK perceives signals from the electron transport chain to link its status to the chloroplast gene expression.

Overall, the paper reports on very interesting observations which remain, however, at the in vitro level. Obtaining evidence for the presence of an Fe-S cluster in CSK in vivo or for changes in the redox state is somehow mandatory to validate the findings.

We thank reviewer 2 for the overall favorable assessment of our manuscript. We agree that the discoveries reported in the current manuscript entail experiments that are conducted mostly in vitro. However, we wish to point out that the characterization of the Chloroplast Sensor Kinase (CSK) at the protein level is a powerful approach to decipher the function of this elusive protein kinase. Furthermore, we demonstrate inhibition of CSK kinase activity by plastoquinol, which by far is the most direct approach to demonstrate the in vivo sensory function of CSK. We additionally provide the midpoint redox potential of CSK Fe-S cluster, a significant advance by itself, which further supports the plastoquinone regulation of CSK. We indeed provide in vivo evidence for serine/threonine kinase activity in *Arabidopsis* CSK, and as specified below we will supply further in vivo data in support of the PQ sensory function of CSK.

I have listed below several other concerns that the authors might consider to make the story more convincing/appealing.

1. I understand from the text (line 86) that full-length proteins from *A. thaliana* and *Phaeodactylum* have been expressed in *E. coli*. Does it mean that the chloroplastic targeting signal has not been removed? This would be a problem as the expressed proteins would not correspond to their physiological mature forms. This may explain why the recombinant *Arabidopsis* CSK has no kinase activity (lines 150-151).

Our inclusion of the targeting peptide of CSK in the recombinant protein was based on the observation that the CSK precursor protein is not processed (Puthiyaveetil et al PNAS, 2008). CSK seems to belong to a rare group of chloroplast proteins whose precursor and mature forms are identical. We believe that the lack of in vitro kinase activity of *Arabidopsis* CSK may be due

to nonnative assay conditions or incorrect folding of the recombinant protein.

2. From the mutagenesis results, it is not clear what are the ligands of the Fe-S cluster. In fact, it is puzzling that the only cysteine, which seems to contribute to the binding (Cys19), is not conserved in orthologs from photosynthetic organisms (see Fig S1).

This brought me several comments :

a. In vitro Fe-S cluster reconstitution may result in artifacts when the reductant (DTT here) is not removed as the thiol groups may serve as ligands. Second, PD10 is often not enough to remove reconstitution reagents and this likely requires an ion exchange chromatography step. Third, in line 338, what means precipitant ?

We agree that the identification of all ligands of the cluster requires further studies. One of the cysteine (Cys 19) is clearly involved as shown by our Fe and S quantification (Table 1). Although some species lack this cysteine, as apparent from the multiple sequence alignment (Fig. S1), there are nearby cysteine residues that may substitute it (Fig. S1). Based on our new x-ray absorption spectroscopy data in the revised manuscript, we now discuss the nature of the other Fe-S ligands as detailed under point below.

We agree with the reviewer that the thiol groups of DTT may act as inadvertent ligands. To examine this possibility, we buffer exchanged reconstituted CSK into a buffer containing DTT using a PD-10 column. Another aliquot of reconstituted CSK was buffer exchanged into a DTT minus buffer using a PD-10 column. A third aliquot was buffer exchanged first into a DTT minus buffer with a PD-10 column, followed by a second purification step involving anion exchange chromatography. The latter analytical technique, as the reviewer suggests, should remove any residual DTT left behind after desalting by the PD-10 column. All three CSK preparations have the same EPR signature of a [3Fe-4S] cluster (Fig. 4a), demonstrating that the DTT does not serve as ligands of the cluster. In line 397 (revised manuscript), by “precipitant” we mean unbound Fe and S.

b. Further spectroscopic analysis (resonance Raman for instance) are required to identify the nature of the ligands.

We agree. To identify the nature of the Fe-S ligands we undertook x-ray absorption spectroscopy (XAS) of Syn-CSK. We chose this technique for its excellent analytical power of metal clusters. The XANES spectra of SYN-CSK shows characteristic long-range atomic interactions (Fig. 4b-c, Table 2, Fig. S2 and S3), indicative of Fe-S and Fe-Fe interactions within the cluster and the Fe-S interaction as arising from the cysteinyl coordination of Fe. It also revealed a short-range Fe-N/O interaction, consistent with the ligation of Fe by the N or O sidechains of histidine or aspartate residues. The cysteine mutagenesis studies (Table 1) together with the XAS data (Table 2) support the existence of non-cysteine ligands for the CSK Fe-S cluster. Unraveling the identity of these amino acid residues will require further high-resolution structural data.

3. The observation that residues possibly forming an amphipathic helix are present in the GAF domain of CSK is interesting but the authors have to follow up this idea and demonstrate whether or not it can promote attachment to thylakoid membranes.

The reviewer makes a good point. We agree that how a presumably soluble CSK protein senses a membrane-intrinsic signal is an intriguing question and that the characterization of the putative amphipathic helix in CSK will be key to answering this. However, this aspect of CSK sensory function is clearly beyond the scope of the current manuscript. The *in vitro* reconstituted liposome binding assays and its optimization with relevant controls, and the *in vivo* mutagenesis studies of the amphipathic helix will take at least 2 years to complete.

4. It is not clear how redox changes on the Fe-S cluster in CSK can translate into changes in gene expression (as proposed in the model) as the kinase activity of CSK seems independent on the presence of the Fe-S cluster (supplementary fig 3).

Our studies suggest that the apo-CSK is highly active as a protein kinase and that the incorporation of the Fe-S cluster (holo-CSK) tames the kinase activity (Fig. S5, revised manuscript). Reduction of the Fe-S cluster by plastoquinol will suppress CSK kinase activity completely. CSK likely occurs only as a holoenzyme *in vivo* as the nascent CSK monomers will promptly incorporate the Fe-S cluster during their biogenesis. The apo-CSK is therefore an experimental artifact and holds no relevance to *in vivo* CSK function.

A separate analogy for CSK function is that the Fe-S cluster acts as a break for CSK kinase activity, with the reduction of the cluster equivalent to fully depressing the break pedal and the oxidation, releasing it. We explain this better in the revised text.

5. With the phosphopeptide enrichment method, do we get information on the level of protein phosphorylation ? This would nice to have such an estimate and correlation with the status of the electron transport chain.

We tried to quantify the level of CSK phosphorylation by spectral counts of the phosphopeptide in PQ oxidizing condition (far-red light) and PQ reducing condition (orange light). Our analysis reveals 2 spectral counts of CSK phosphopeptide under far-red light in two separate experiments, while the orange light retrieved none (Table 3, revised manuscript). We have further tried to quantify the extent of CSK autophosphorylation by the phos-tag method, which separates phosphorylated proteins from unphosphorylated proteins by gel electrophoresis. Unfortunately, the phos-tag method was unable to resolve any phosphorylated CSK proteins in the samples we analyzed. It is likely that this is due to very low level of phosphorylation, as apparent from the mass spectrometry results (Table 3). Deciphering the *in vivo* phosphorylation dynamics of CSK will require additional extensive future studies.

By the way, in line 357 : give details or reference again (as in line 158) on how phosphopeptide enrichment was performed.

This reference is now given as number 17 in the bibliography.

Minor comments :

Line 172 : redox reagents
Line 330 NaCl
Line 353 : phosphopeptide
Line 725 : I would suggest to remove « by air ».

Thank you. All minor comments are addressed in the revised manuscript.

Reviewer #3 (Remarks to the Author):

The chloroplast protein kinase CSK was identified more than 10 years ago. Whether this kinase plays an important role in adjusting chloroplast gene expression in response to changes in the redox state of the photosynthetic electron transport chain remains a controversial issue and its main role is still unclear.

We thank the reviewer for the evaluation of our manuscript. We agree that the full functional significance of CSK is yet to be unraveled. However, we wish to point out that the photosystem I (PS I) gene regulatory role of CSK has been unambiguously demonstrated in the original study (Puthiyaveetil et al PNAS, 2008). Observations on the cyanobacterial CSK homologue, hik2, further supports the photosynthesis gene regulatory role of CSK (Ibrahim et al Frontiers in Plant Science, 2016). CSK binds a quinone analogue with high affinity (Puthiyaveetil et al 2013), shares a gene expression phenotype with sigma factor 1 (SIG1) phosphorylation site mutant (Shimizu et al PNAS 2010), and interacts with SIG1 (Puthiyaveetil et al GBE, 2010). These evidences point to plant CSK acting as a sigma factor kinase, in relaying plastoquinone redox signal to PS I gene expression machinery.

In this manuscript the authors have performed an extensive and thorough biochemical analysis of CSK from different photosynthetic organisms and they show that it contains a 3Fe-4S center and that its autocatalytic kinase activity is activated under oxidizing conditions and inactivated under reducing conditions. This is clearly an important finding but this study does not provide further clues on the functional role of this kinase.

We thank the reviewer for this assessment. As the reviewer notes we have approached the CSK problem from a different vantage point, i.e. from the point of the sensory mechanism of CSK. In the current manuscript we provide rigorous evidence for a redox responsive [3Fe-4S] cluster in CSK. We reveal the redox potential of the CSK cluster, which shows a one electron redox behavior. In the revised manuscript, we provide new x-ray absorption data on CSK Fe-S cluster with information on the nature of its ligands. We further show CSK autokinase regulation by plastoquinone. This is an extensive characterization of the upstream signaling events in CSK-based two-component system, shedding critical insights into its light acclimatory function.

In the revised manuscript we further provide evidence for CSK-genetic control on non-photosystem gene expression (Fig. 9). Although this is not an extensive analysis of all possible CSK gene targets, it is a clear indication that CSK-genetic control goes beyond PS I genes so as

to regulate the stoichiometry of both photosystems and non-photosystem electron transport complexes in chloroplasts. A thorough characterization of CSK's chloroplast gene regulatory function will be the subject of future research from our laboratory and elsewhere.

Additional comments

Statements such as a "PQ-controlled photosystem gene expression forms the basis of the photosystem stoichiometry adjustment" (l. 63-64) are controversial as most of the current literature does not support such a claim.

We agree with the reviewer that the photosystem stoichiometry adjustment may involve multiple regulatory pathways and that the PQ-control may just be one facet of this acclimatory response. We have rephrased this sentence accordingly in the revised manuscript (line 64).

Table 2 lacks a control of the kinase activity under reducing conditions

Agree. This is now addressed. Please see response to point 5 of reviewer 2

Reviewers' comments:

Reviewer #1 (Remarks to the Author):

This is a thorough revision that in my view takes into account all of the criticisms raised by referees in the first round. They present extensive new X-ray absorption spectroscopy data that further support their case. The modifications to the text accommodate the suggestions of the referees. The paper makes a very substantial contribution to understanding the sensing of environmental signals and their transduction into gene regulation in plastids. The paper should move forward.

Reviewer #2 (Remarks to the Author):

In this revised version, the authors have clarified some questions and substantially improved the manuscript. I understand that not all my concerns could be addressed, but I still have two comments.

1. Concerning my former comment 2a : it seems that my expectations were not enough clear, sorry for that. I was in fact recommending (for reasons already detailed) to perform the Fe-S cluster reconstitution experiments in the absence of DTT i.e. using pre-reduced and DTT-desalted proteins, not to remove DTT afterwards. I should say that answering this point is less relevant now that EXAFS data indicating the presence of non-sulfur ligands have been added. Hence, the new figure 4a has no more reason to exist at least as a main figure.

On this aspect of Fe-S cluster ligands, I would also moderate the statement made on line 125 « two cysteine residues in CSK that are highly conserved from cyanobacteria to higher plants ». In fact, as already pointed out in my previous comment, these cysteines are absent in a number of sequences and I would not use highly (that is moreover not really quantifiable) to describe such a conservation.

2. Concerning my former comment 3. I was in fact thinking to a simpler experiment showing whether CSK associates or not (or at least partially) with thylakoid membranes. Having this experiments would considerably strengthen the model and conclusions.

Minor point : p123 replace sulfur by sulfure

Reviewer #3 (Remarks to the Author):

The authors have improved their manuscript. However I have still doubts about the role of CSK in chloroplast transcriptional control. First, the references mentioned by the authors as providing experimental support for such a role are not convincing and to the best of my knowledge have not been repeated by other laboratories. Second, the new transcriptional data presented in Fig. 9 show only examples of genes whose expression is increased in the *csk* mutant and decreased when CSK is overexpressed. However it is essential that the authors also examine the transcription of PSII genes such as *psbA* and *psbD* which should show the opposite behavior if indeed CSK plays an important role in redox transcriptional control. In the absence of this control it is difficult to assess the significance of the results shown in fig. 9.

Reviewers' comments:

Reviewer #1 (Remarks to the Author):

This is a thorough revision that in my view takes into account all of the criticisms raised by referees in the first round. They present extensive new X-ray absorption spectroscopy data that further support their case. The modifications to the text accommodate the suggestions of the referees. The paper makes a very substantial contribution to understanding the sensing of environmental signals and their transduction into gene regulation in plastids. The paper should move forward.

We thank the reviewer for the positive evaluation of our manuscript.

Reviewer #2 (Remarks to the Author):

In this revised version, the authors have clarified some questions and substantially improved the manuscript. I understand that not all my concerns could be addressed, but I still have two comments.

1. Concerning my former comment 2a : it seems that my expectations were not enough clear, sorry for that. I was in fact recommending (for reasons already detailed) to perform the Fe-S cluster reconstitution experiments in the absence of DTT i.e. using pre-reduced and DTT-desalted proteins, not to remove DTT afterwards. I should say that answering this point is less relevant now that EXAFS data indicating the presence of non-sulfur ligands have been added. Hence, the new figure 4a has no more reason to exist at least as a main figure. On this aspect of Fe-S cluster ligands, I would also moderate the statement made on line 125 « two cysteine residues in CSK that are highly conserved from cyanobacteria to higher plants ». In fact, as already pointed out in my previous comment, these cysteines are absent in a number of sequences and I would not use highly (that is moreover not really quantifiable) to describe such a conservation.

We thank the reviewer for the clarification. If possible, we wish to keep Figure 4a as a main figure. We feel that Figure 4a with the other two panels comprehensively test the nature of the Fe-S ligands in CSK as initially requested by the reviewer. On reviewer's point about moderating the statement on conservation of the cysteine residues, we have now removed the word "highly" as suggested.

2. Concerning my former comment 3. I was in fact thinking to a simpler experiment showing whether CSK associates or not (or at least partially) with thylakoid membranes. Having this experiments would considerably strengthen the model and conclusions.

Thank you for this suggestion. Using a dot immunoblotting technique we now provide evidence for thylakoid membrane association by CSK as requested by the reviewer (Supplementary Fig. 4a). Further examination of the role of the CSK amphipathic helix in membrane interaction will require extensive in vitro and in vivo assays, which are clearly beyond the scope of the current manuscript as stated in the earlier response letter.

Minor point : p123 replace sufur by sulfur

Thank you. Corrected.

Reviewer #3 (Remarks to the Author):

The authors have improved their manuscript. However I have still doubts about the role of CSK in chloroplast transcriptional control. First, the references mentioned by the authors as providing experimental support for such a role are not convincing and to the best of my knowledge have not been repeated by other laboratories. Second, the new transcriptional data presented in Fig. 9 show only examples of genes whose expression is increased in the *csk* mutant and decreased when CSK is overexpressed. However it is essential that the authors also examine the transcription of PSII genes such as *psbA* and *psbD* which should show the opposite behavior if indeed CSK plays an important role in redox transcriptional control. In the absence of this control it is difficult to assess the significance of the results shown in fig. 9.

Thank you for the suggestion. As requested by the reviewer, we now provide gene expression data for *psbA* and *psbD* (Fig. 9). These PS II genes do not show any statistically significant changes, consistent with CSK's role in bringing about photosystem stoichiometry changes through regulation of only PS I genes. Adjustment of photosystem stoichiometry through changes in only PS I abundance is a recurring theme from cyanobacteria to plants (See for example, Fujita et al, *Plant Cell Physiol* 1988, Melis et al, *Photosynthesis Research* 1996, and Mulo et al, *BBA Bioenergetics* 2012).

REVIEWERS' COMMENTS:

Reviewer #2 (Remarks to the Author):

I thank the authors for the efforts in answering all the points, even though I believe that the experiments shown in figure S4 could have been executed in a better and more convincing manner.

Reviewer #3 (Remarks to the Author):

In their revised version, the authors have performed the control experiment I requested and shown that the PSII genes respond in a different way from the PSI genes in the absence of CSK and when this kinase is overexpressed. This significantly strengthens the manuscript.